# Prognostic Indicators of Overall Survival in Hepatocellular Carcinoma Patients Undergoing Liver Resection

**DOI:** 10.3390/cancers16071427

**Published:** 2024-04-07

**Authors:** Cristina-Paula Ursu, Andra Ciocan, Ștefan Ursu, Răzvan Alexandru Ciocan, Claudia Diana Gherman, Ariana-Anamaria Cordoș, Dan Vălean, Rodica Sorina Pop, Luminița Elena Furcea, Bogdan Procopeț, Horia Ștefănescu, Emil Ioan Moiș, Nadim Al Hajjar, Florin Graur

**Affiliations:** 1Department of Surgery, “Iuliu Hațieganu” University of Medicine and Pharmacy, Croitorilor Street, No. 19–21, 400162 Cluj-Napoca, Romania; cristinapaulapop10@yahoo.com (C.-P.U.); stefan.ursu20@gmail.com (Ș.U.); valean.d92@gmail.com (D.V.); luminita.furcea@umfcluj.ro (L.E.F.); drmoisemil@elearn.umfcluj.ro (E.I.M.); nadim.alhajjar@umfcluj.ro (N.A.H.); florin.graur@umfcluj.ro (F.G.); 2“Prof. Dr. Octavian Fodor” Regional Institute of Gastroenterology and Hepatology, Croitorilor Street, No. 19–21, 400162 Cluj-Napoca, Romania; bogdan.procopet@umfcluj.ro (B.P.);; 3Department of Surgery-Practical Abilities, “Iuliu Hațieganu” University of Medicine and Pharmacy, Marinescu Street, No. 23, 400337 Cluj-Napoca, Romania; razvan.ciocan@umfcluj.ro (R.A.C.); gherman.claudia@umfcluj.ro (C.D.G.); cordos.ariana@umfcluj.ro (A.-A.C.); 4Romanian Society of Medical Informatics, 300041 Timișoara, Romania; 5Department of Community Medicine, “Iuliu Hațieganu” University of Medicine and Pharmacy, Avram Iancu Street, No. 31, 400347 Cluj-Napoca, Romania; drsorinapop@yahoo.com; 6Department of Internal Medicine, “Iuliu Hațieganu” University of Medicine and Pharmacy, Croitorilor Street, No. 19–21, 400162 Cluj-Napoca, Romania

**Keywords:** hepatocellular carcinoma, hepatic venous pressure gradient, liver stiffness, tumor burden score, liver resection

## Abstract

**Simple Summary:**

Hepatocellular carcinoma (HCC) ranks as the third leading cause of cancer-related deaths worldwide, with liver resection being the most effective for curative intent. The hepatic venous pressure gradient (HVPG), transient elastography-liver stiffness measurement (TE-LSM), and TAC score are intricately connected with the postoperative evolution of cirrhosis. The primary objective of our study is to evaluate the predictive value of key parameters for surgical patients. The proposed predictors can better presume treatment outcomes in HCC and potentially allow an improvement in therapeutic strategy.

**Abstract:**

Hepatocellular carcinoma (HCC) is the predominant form of primary liver cancer and the third contributor to malignancy-related deaths worldwide. The hepatic venous pressure gradient (HVPG), transient elastography-liver stiffness measurement (TE-LSM), and the association between TBS (tumor burden score), alpha-fetoprotein levels, and the Child–Pugh classification (TAC score) can serve as valuable prognostic indicators for these patients. Therefore, the main objective of our research was to analyze the prognostic value of the HVPG, TE-LSM, TBS, and TAC scores. An observational and survival study was conducted on 144 subjects. Our findings indicated that HVPG greater than 10 mmHg, AFP surpassing 400 ng/mL, an advanced C–P class, and low TAC score are independent predictors of overall survival. During the multivariate analysis, AFP serum levels and C–P class proved statistically significant. The present study revealed significant differences in overall survival between the two groups divided upon HVPG values and settled by the cutoff of 10 mmHg (*p* = 0.02). Moreover, by dividing the cohort into three groups based on the TAC score (very low, low, and moderate), statistically significant differences in overall survival were observed across the groups (*p* = 0.004).

## 1. Introduction

Hepatocellular carcinoma (HCC) is the most frequent form of primary liver malignancy and ranks as the third leading cause of cancer-related mortality on a global scale [1,2]. A significant number of cases arise in individuals with chronic liver disease, notably linked to hepatitis B virus (HBV) or hepatitis C virus (HCV) infections, as well as those with a history of chronic alcohol consumption [1,2,3,4]. Discrepancies in the incidence of HCC are closely tied to the prevalence of chronic hepatopathies [1,5,6,7]. Regular evaluation and screening for HCC in cirrhotic patients is crucial [8,9,10]. Current advances in early diagnosis and therapeutic interventions have improved prognosis and overall survival. Unfortunately, hepatocellular carcinoma patients are typically diagnosed at an intermediate or advanced stage. Therefore, the selection of the appropriate treatment approach is based on tumor stage, liver function, and performance status [2,5,6,11,12], according to the revised Barcelona Clinic Liver Cancer (BCLC) staging system [9,11,12]. The treatment strategies proposed by BCLC are heavily influenced by liver function in patients with significant underlying disease and encompass liver resection (LR), radiofrequency or microwaveablation (RFA/ MWA) and liver transplantation (LT) for early stages (0,A), transarterial chemoembolization (TACE) and liver transplantation for the intermediate stages (B), and systemic chemotherapy (C) and supportive care (D) for advanced stages [11,12]. Surgical resection provides the best results in terms of overall and disease-free survival for individuals with very early or early HCC and preserved liver function. Despite multiple therapeutic options, the early diagnosis of hepatocellular carcinoma remains challenging, therefore leading to a poor prognosis and high mortality [11,12,13,14].

Portal hypertension is the result of liver fibrosis due to cirrhosis. As portal hypertension becomes clinically significant, it can lead to complications like hemorrhagic esophageal varices, ascites, spontaneous bacterial peritonitis, and hepatorenal syndrome. Therefore, it is crucial to assess the grade of portal hypertension prior to treatment [15,16]. The gold standard method for the evaluation of portal hypertension is the angiographic measurement of the hepatic venous pressure gradient (HVPG), resulting from the difference between the portal and the systemic pressure assessed in the hepatic outflow system. The normal range for HVPG is 0–5 mmHg; values above 10 mmHg demonstrate clinically significant portal hypertension, while values above 15 mmHg are prohibitive for hepatectomies [17,18,19] due to the increased likelihood of postoperative decompensation and high mortality [17,18,20,21]. Alterations in liver microcirculation play a crucial role in chronic liver disease and HCC, whether it is considered from the perspective of portal hypertension, porto-sinusoidal vascular disease, or immunological responses. Research in murine models reveals that when HBV-specific CD8+ T cells receive inhibitory signals from HBV-expressing hepatocytes, defective cells expand [22]. Active CD8+ T lymphocytes strongly engage with platelets to simulate hepatic blood flow, thus suggesting that a physical platelet–T cell interaction leads to the hepatic accumulation of the latter cells. Inadequate CD8+ T cell response fails to eliminate HBV from the liver, leading to low-level hepatocellular damage, liver fibrosis, cirrhosis, and HCC as a consequence. Therefore, a therapeutic approach that reduces CD8+ T cell liver damage and prevents or delays fibrosis could be efficient in hepatocellular carcinoma [22,23].

Transient elastography (TE) is an ultrasonography-based technique that measures liver stiffness (referred to as TE-LSM) on a kilopascal scale, which enables chronic liver disease grading according to the Fibroscan class (Metavir 0 to 4) [15,17,20,21]. TE-LSM has a strong association with HVPG, hence serving as a viable non-invasive method for assessing liver fibrosis and cirrhosis (above 12–14 kPa), the normal range being between 4 and 6 kPa. Multiple studies have explored the association between TE-LSM and HVPG, as well as the diagnostic efficacy of TE-LSM in detecting CSPH and its potential as an alternative to HVPG [15,17,20,21,24,25,26,27], but the major drawback is its inconsistency when liver masses appear. The integration of HVPG with liver stiffness measurements can offer important insights in determining the optimal therapy for patients with HCC [17,20,21,24,25,26,27,28,29].

The tumor burden score (TBS) is an innovative scoring system that combines both the number and the size of tumors. Although originally designed for patients undergoing liver resections for colorectal metastases, this score has lately been effectively applied in evaluating HCC patients receiving surgical treatment. As previously stated, the heterogeneity of HCC treatment options is linked to differences in tumor morphology, tumor biology, and liver function. Therefore, we sought to stratify the prognosis of these patients due to a few factors that impact the choice of therapy and outcome of hepatocellular carcinoma patients: liver stiffness, hepatic venous pressure gradient, tumor burden score, serum AFP (alpha-fetoprotein) and Child–Pugh (CP) classification. The latter three variables are encompassed under the TAC score, a recently developed tool by Lima et al., which is used to evaluate long-term prognosis following resection for HCC patients [8].

The aim of our research was to evaluate the aforementioned prognostic factors in association with postoperative decompensation and survival in patients undergoing surgical therapy for hepatocellular carcinoma.

## 2. Materials and Methods

### 2.1. Study Design

The present paper is a retrospective observational and survival study conducted between January 2015 and December 2022 on 144 patients diagnosed with hepatocellular carcinoma admitted to the Surgical Department of “Prof. Dr. Octavian Fodor” Regional Institute of Gastroenterology and Hepatology, Cluj-Napoca, Romania.

A comprehensive analysis was conducted on the medical records of individuals who had a conclusive histopathological diagnosis of hepatocellular carcinoma. Demographic information (e.g., gender, age, environment) was collected. Clinical and paraclinical examinations (e.g., the presence of hepatitis B or C virus infection, chronic alcohol intake, steatohepatitis, chronic liver disease, total and conjugated bilirubin, albumin, total proteins, thrombocytes, INR, AFP, GOT (glutamic-oxaloacetic transaminase) and GPT (glutamic-pyruvic transaminase), ascites, encephalopathy. TE-LSM and HVPG were evaluated, and C–P classification, TBS, and BCLC staging were assessed at admission prior to therapy allocation. Operative details, including the type of surgical procedure, the duration of intervention, intraoperative blood loss, and histopathological characteristics of the resected specimen (e.g., number of nodules, dimension), were introduced. Further, postoperative data regarding follow-up (postoperative liver decompensation and survival) were obtained. Patients treated with curative intent were followed up at 3, 6, and 12 months in the first year.

### 2.2. Inclusion and Exclusion Criteria

The following inclusion criteria were taken into consideration to define the population analyzed while performing this research:
(a)A final histopathological diagnosis of hepatocellular carcinoma;(b)Patients who underwent surgical treatment with curative intent;(c)Subjects with liver function graded Child–Pugh A or B.


Individuals with a different histopathological diagnosis, with incomplete or missing data regarding hepatocellular carcinoma, and patients who did not benefit from surgery during their hospitalization period were excluded from the study.

### 2.3. HVPG and TE-LSM Measurements

HVPG measurements were realized before the surgery, under local anesthesia and radiologic guidance by specialized operators. A 9F venous catheter was inserted into the right internal jugular vein using the Seldinger technique. Subsequently, a 7F balloon-tipped catheter was advanced into the right hepatic vein. HVPG was calculated as the difference between wedged and free hepatic venous pressures. The threshold for clinically significant portal hypertension was established as an HVPG value equal to or greater than 10 mmHg.

Liver stiffness was measured after 8h of fasting before surgery. The probe was placed on the right lobe of the liver through intercostal spaces while the patient adopted a supine position, with the right arm placed in maximal abduction. A total of ten valid measurements were performed for each individual patient. LSM was expressed in kilopascal (kPa).

With the aim of evaluating the overall survival of subjects diagnosed with HCC who underwent surgical treatment, we divided them into groups based on thresholds for HVPG: values under 10 mmH and greater than 10 mmHg, and for TE-LSM, the cut-off values considered were <10 kPa, 10–25 kPa and >25 kPa. A value of LSM < 10 kPa, when other clinical or imaging indications were not present, ruled out compensated advanced chronic liver disease (cACLD), and values >15 kPa strongly indicated cACLD. A score equal to or higher than 25 kPa was deemed to be sufficient to rule in CSPH, defining the group at a greater probability of decompensation [17]. Suitable measures were limited to those that achieved a success rate of 60% and had an interquartile range to median ratio of less than 30%. The operator was unaware of the patient’s clinical data.

### 2.4. Tumor Burden Score and TAC Score

TBS is obtained using the following formula: TBS^2^ = (maximum tumor diameter in cm)^2^ + (number of tumors)^2^. TBS solely necessitates the diameter of the largest tumor and the number of malignant nodules. Patients were divided into three groups in accordance with the previous descriptions of this score: low TBS values were considered less than 3.36, medium TBS values were taken between 3.36 and 13.74, and high TBS was over 13.74. As for estimating the TAC score, the following system was used: TBS low/medium/high = 0/1/2 points, AFP defined as low < 400 ng/mL, high > 400 ng/mL with 0/1 point, and C–P classification A/B = 0/1 point, respectively. Patients were classified on a scale ranging from 0 to 4 and divided into the following scoring groups: very low, low, medium, and high [8].

### 2.5. Surgical Resection

The appropriate course of treatment was determined by a multidisciplinary team and operated by surgeons with confirmed expertise in liver surgery. Liver resections (LRs) were categorized into two groups: minor liver resections, which involved the removal of up to two segments, and major liver resections. Major liver resections encompassed right and left hepatectomies, extended right and left hepatectomies, as well as liver resections involving three or more segments. The choice of procedure depended on specific factors such as tumor localization, the number of nodules, and size. The Brisbane 2000 classification was employed for nomenclature considerations in defining the terminology associated with various types of LR [30,31]. Furthermore, the treatment was divided into anatomical and non-anatomical approaches based on the systematic removal of a hepatic section confined by lesion-bearing portal tributaries.

### 2.6. Research Endpoints

To conduct the present research, we employed the following endpoints: postoperative decompensation and mortality. Postoperative decompensation was defined as the presence of one of the following during the patients’ follow-up: liver failure, jaundice (described as hyperbilirubinemia >3 mg/dL), hepatic encefalopathy, ascites, posthepatectomy hemorrhage, as well as acute kidney injury. Mortality was defined as the occurrence of death during the first 90 days of follow-up.

### 2.7. Statistical Analysis

Statistical analysis and graphical generation were conducted utilizing SPSS (IBM, version 26). The normality of data was tested using the Shapiro–Wilk and Kolmogorov–Smirnov tests. Continuous and normally distributed variables were interpreted using the *t*-test for independent values. The association between qualitative values was evaluated using Pearson’s Chi-squared tests. In order to evaluate overall survival, Kaplan–Meier curves were used and evaluated with the Log-Rank test. Univariate analysis and multivariate analysis were performed using Cox proportional hazards. The significance value was considered at *p* < 0.05.

Based on the results of the survival analysis of our data, we simplified it into HVPG < 10 mmHg and >10 mmHg; TE-LSM results were <10 kPa, between 10 and 25 kPa and >25 kPa.

### 2.8. Ethical Approval

The study was approved by the ethical department of “Prof. Dr. Octavian Fodor” Regional Institute of Gastroenterology and Hepatology, Cluj-Napoca, Romania (15602/22 December 2022) and “Iuliu Hațieganu” University of Medicine and Pharmacy, Cluj-Napoca, Romania (No. 24/13 February 2023). In each situation, informed consent was obtained in accordance with the principles outlined in the Declaration of Helsinki.

## 3. Results

### 3.1. Descriptive Characteristics of Enrolled Patients

The baseline clinical and paraclinical characteristics of our cohort are included in Table 1. The patients enrolled in our research had a mean age of 66 years old, with 65.98% of the patients pertaining to the male sex. The paraclinical examinations showed a total bilirubin mean value of 0.9 mg/dL with a standard deviation of 0.60450% and a mean value for albumin of 4.059 g/dL. The mean MELD score was 9.16 (±2.58). For the factor included in the TAC score, our cohort had a mean value for TBS of 4.78 and 102 ng/mL for AFP. The mean value for INR was 1.203. The mean value of the HVPG measurement was 9.11 mmHg, while the mean value for TE-LSM was 16.59 kPa. The cause of liver cirrhosis was alcohol consumption in 19.4% of patients, chronic hepatitis C in 45.1%, B in 18.1%, and idiopathic in 17.4%. Encephalopathy occurred in two cases (1.2%), and 92.2% of the patients had no ascites before the surgical treatment, demonstrating compensated underlying liver disease. Concerning the C–P classification, 70.8% of the patients were C–P A, while the rest of them were C–P B class. Patients were divided according to the BCLC 2022 staging system as follows: 0 or very early stage (5.6%), A or early stage (78.5%), B or intermediate stage (13.9%), and C or advanced stage (2.1%). Most of the surgical interventions were non-anatomic, tissue-sparing (69.4%). Among the 144 patients included in our research, 22.2% experienced postoperative decompensation, with nine of them specifically developing posthepatectomy liver failure (PHLF). Additional forms of hepatic decompensation observed in our study included ascites, hyperbilirubinemia, encephalopathy, acute kidney injury, postoperative hemorrhage, and prolonged INR. Regarding tumoral recurrence, 20.1% of patients reported a single recurrence during their follow-up period, whereas 21.5% had multiple recurrences. Among the 62 patients who succumbed, 32 of them had clinical or imagistic confirmed recurrences.

When examining the relationships between the TE-LSM, HVPG, TBS, and MELD score with postoperative decompensation and survival for up to 12 months, no statistically significant differences were found, except for the average value of the MELD score. MELD was significantly different between those who experienced postoperative decompensation (8.39 ± 2.34) and those who did not (10.06 ± 3.19), with a *p*-value of 0.03. Similarly, significant differences between the values corresponded to the 3-month survival: 8.99 ± 2.39 vs. 11.67 ± 3.96, *p* = 0.002 (Table 2).

### 3.2. Association of HVPG, TE-LSM and TAC Score with Overall Survival

When analyzing the Child–Pugh class, there were statistically significant disparities in terms of overall survival between Child A and Child B classes (62.7% vs. 11.1% OS, 62.17 (+/−3.95) vs. 26.9 (+/−6.04) months, *p* < 0.001). Statistically significant differences were observed when establishing HVPG cutoff values (less than 10 mmHg vs. more than 10 mmHg: 66.7% vs. 48.4% OS, 58.06 (+/−4.16) vs. 39.42 (+/−4.49) months, *p* = 0.02), AFP score (under 400ng/mL vs. over 400ng/mL, 65.0% vs. 28.6 OS, 62.93 (+/−4.06) vs. 32.99 (+/−8.42) months, *p* = 0.003) and TAC score (very low vs. low vs. intermediate: 72.1% vs. 61.3% vs. 21.4% OS, 69.12 (+/−6.12) vs. 55.51 (+/−4.82) vs. 33.42 (+/−8.44) months, (*p* = 0.004) (Figure 1). There were no statistically significant differences regarding TE-LSM score (less than 10 kPa vs. 10–25 kPa vs. over 25 kPa), 67.9% vs. 51.2% vs. 61.5% OS, 56.07 (+/−5.39) vs. 49.41 (+/−4.95) vs. 69.08 (+/−8.89) months, *p* = 0.36) and tumor burden score (low vs. intermediate vs. high, 61.7% vs. 53.1% vs. 66.7% OS, 59.68 (+/−5.43) vs. 53.05 (+/−4.02) vs. 46.00 (+/−12.24) months, *p* = 0.75) (Figure 2).

### 3.3. Preoperative Factors Associated with Overall Survival

Proportional hazards were evaluated individually using Cox regression. (Table 3) Accounting for univariate analysis, we found statistically significant differences in terms of HVPG higher than 10 mmHg (HR = 2.08, CI—1.07–4.06, *p* = 0.03), AFP over 400 ng/mL (HR = 2.77, CI—1.37–5.62, *p* = 0.004), Child–Pugh classification (HR =3.31, CI—1.84–5.95, *p* < 0.001) and TAC score (low score, HR = 2.98, CI—0.91–9.87, *p* = 0.03; intermediate score HR = 2.74, CI—0.83–9.03, *p* = 0.04).

## 4. Discussion

Hepatocellular carcinoma is characterized by heterogeneity and complexity [12]. Frequently, patients diagnosed with HCC present with underlying cirrhosis, thereby necessitating the assessment of the hepatic venous pressure gradient and liver stiffness measurement as crucial indicators for evaluating the extent of liver disease. These parameters play a significant role in guiding the therapeutical approach for patients [12,15,18,21]. Furthermore, the allocation of therapies for patients with hepatocellular carcinoma needs to consider tumor morphology and liver function. As a result, staging becomes an essential requirement, and the evaluation of both HVPG and LS provides an in-depth overview of the outcomes associated with various therapeutic interventions [9,12,25]. When considering serum indicators, elevated levels of alpha-fetoprotein are linked to a poor prognosis. Increased values of alpha-fetoprotein can predict tumor recurrence following resection, the level of response to loco-regional therapy, the drop-out risk among patients awaiting liver transplantation (LT), as well as survival and tumor recurrence following LT, and survival in advanced HCC. Usually, AFP levels exceeding 200 or 400 ng/mL can be regarded as prognostic indicators of unfavorable outcomes. [8,12,32]. Consequently, our study investigated the levels of AFP, showing a mean value of 102 ng/mL (±298.81). It is worth noting that the use of AFP has primarily been evaluated in the context of diagnostic applications rather than for surveillance purposes because it demonstrates inadequate performance [12,33]. With respect to the presence of cirrhosis as an underlying condition, most authors evaluated patients classified as Child–Pugh A since it describes a group of individuals with preserved liver function and appropriate candidates for liver resection [19,34,35,36]. The TAC score was designed using criteria that are simple to calculate and commonly examined in the clinical context. TAC evaluated tumor morphology, biology, and liver function by combining TBS, AFP, and CP. TBS has been verified as a useful method for summarizing the total tumor extent, and it has proven to be a strong predictor of outcomes following HCC resection. Despite this, TBS is not extensively used in HCC prognostic models. Lima et al. demonstrated a significant correlation between a higher TAC score and unfavorable clinicopathological characteristics, such as advanced T disease, microvascular and lymphovascular invasion, and poor-to-undifferentiated tumor burden (all *p* < 0.001). In addition, there was a clear correlation between the TAC score and recurrence patterns, with the recurrence-free survival (RFS) progressively worsening as the TAC values increased. The TAC score showed superior performance in comparison to other commonly utilized models, including CLIP, JIS, AJCC staging, and the BCLC. TAC exhibited higher prognostic efficacy compared to all the previously mentioned staging systems following liver resection for HCC [8,37,38,39].

This research intended to supplement the existing information in the field of hepatocellular carcinoma by investigating the utility of the hepatic venous pressure gradient, liver stiffness measurements, tumor burden score, and TAC score, respectively, as prognostic indicators for patients undergoing surgical procedures. Consequently, data were collected from a cohort of 144 individuals who were diagnosed with HCC over a period of seven years with the purpose of evaluating the relationship between these variables and the overall survival rate. Regarding patients’ demographics, our cohort was characterized by a mean age of 66.17 (±7.44) years old, with most of them pertaining to the male gender. The patient cohort is heterogenous, encompassing individuals with cirrhosis due to chronic hepatitis infection (B: 18.1% and C: 45.1%) and alcoholic liver disease (19.4%). Cucchetti et al. gathered data from 70 patients enrolled in a prospective manner, with a median age of 62, while Azoulay et al. included in their research 79 patients with a median age of 65 [19,34]. Certain published papers present multicenter studies that encompass a substantial cohort of patients, such as the BRIDGE study, and a multiregional longitudinal cohort analysis of newly diagnosed cases of HCC, consisting of a total of 8656 subjects or single-centered analysis on large databases, such as Siu-Ting Lau et al.’s study [4,24,40]. With respect to the presence of cirrhosis as an underlying condition, most authors evaluated patients classified as Child–Pugh A since it describes a group of individuals with preserved liver function and appropriate candidates for liver resection [19,34,35,36]. As a result of C–P A’s limited predictive capability and ongoing efforts to broaden resection criteria [12,19], our cohort included patients pertaining mostly to C–P A and B. Tumor burden score is a recently developed measure for patients with hepatocellular carcinoma. While its usefulness in predicting prognosis has been established, our findings indicate that the most significant factors influencing the TAC score are the AFP serum levels and, more importantly, the C–P classification. Tsilimigras et al. noticed the synergistic impact of AFP and TBS in categorizing patients with HCC. Lima et al. incorporated the C–P classification into the equation, as C–P is the most widely utilized measure for assessing liver function reserve [8,37].

There are several significant studies debating either the importance of HVPG or LSM in the assessment of patients with HCC or the use of both variables to evaluate treatment allocation, postoperative complications, recurrence, and overall outcomes [18,19,20,34,36,41]. Numerous centers rely on several biological scores rather than venous pressure gradient measurements to ascertain the presence and severity of portal hypertension (PH) [23,25,26,42,43]. The occurrence of CSPH, as indicated by an HVPG equal to or greater than 10 mmHg or by the presence of clinical signs of portal hypertension, is linked to an increased likelihood of decompensation and mortality in individuals with cirrhosis who undergo liver resection [12,17,18]. Boleslawski et al. enrolled 43 patients with cirrhosis and resectable HCC to determine whether HVPG measurement and other portal hypertension criteria prior to surgical treatment can be used to predict the postoperative course of these patients. Postoperative liver dysfunction and 90-day mortality constituted the primary endpoints. As a result, preoperative HVPG was associated with postoperative liver dysfunction, and patients with an HVPG of 10 mmHg or surpassing this limit had a lower 3-month survival [44]. Taking this into consideration, a value of HVPG of 10–12 mmHg is helpful in discerning patients who are at risk for in-hospital death [45]. Cirrhotic patients with HCC and CSPH, according to HVPG measurements, may undergo liver resection with acceptable mortality and morbidity rates with manageable acute liver failure, even achieving a favorable outcome under suitable treatment. The attainment of these outcomes is feasible in a subset of patients who possess good hepatic function, favorable overall health, and an adequate future liver remnant [12,40].

In light of the ongoing debate upon the cut-off values predicting unfavorable outcomes for HCC patients with underlying cirrhosis and different degrees of portal hypertension that may benefit from surgical interventions, we tried to identify whether the subjects included had a statistically significant association between values of HVPG and TE-LSM within the preventive limits and postoperative decompensation and survival [12,17,18,40]. There was no significant finding between the HVPG and TE-LSM values under or an HVPG over 10 mmHg regarding either postoperative decompensation or survival at 3, 6, or 12 months (Table 2). Nevertheless, when analyzing threshold values with overall survival, patients with an HVPG under 10mm Hg had a statistically significantly better OS than patients with values over 10 mmHg (*p* = 0.027). In that regard, our research adheres to other studies and Baveno VII norms [17]. However, it is indisputable that in a heterogenous cohort of individuals, some will have values that exceed limits.

One noteworthy observation regarding HVPG is that patients had a mean value ≤10 mmHg, considering that it was a criterion describing the “ideal” candidate for LR in cirrhosis by the European Association for the Study of the Liver (EASL)/European Organisation for Research and Treatment of Cancer (EORTC) Clinical Practice Guidelines in 2012 [12]. Nevertheless, in recent years, selected patients exceeding the above-mentioned criteria have benefited from LR. This approach has been implemented in specialized, high-volume centers, similar to ours, where there is careful consideration of each predictor of prognosis [12,18,34,40].

Using the gold standard method to evaluate CSPH resulted in a substantially strong association between the presence of such hypertension and clinical outcomes, according to Berzigotti et al. [18]. The survival rates at 1- and 3-year intervals for patients with an HVPG less than 10mmHg were both 100%, whereas patients with an HVPG greater than or equal to 10mmHg had survival rates of 97.1% and 79.4% at 1- and 3-year intervals, respectively [19]. Portal hypertension, which is associated with a poorer prognosis and indicates a more advanced evolutionary stage of cirrhosis, should not be regarded as an unforeseen predictor of survival and should be considered in treatment decisions for patients with technically resectable HCCs [18,40]. Alternative treatments to surgery for patients with CSPH include ablation and transplantation, both of which have the potential to achieve long-term survival [12]. The findings of Cucchetti et al. provide confirmation that HVPG ≥ 10mmHg is linked to an increased likelihood of ascitic decompensation in the immediate period after surgery. However, the overly stringent implementation of this threshold disregards a substantial portion of potential surgical candidates, as their liver function tests did not exhibit significant deterioration three months post-surgery [18,19]. HVPG over 10 mmHg remains an independent predictive factor of overall survival. On the other hand, the utilization of HVPG measurements can be employed to adjust the planning process for a hepatectomy, thereby preventing an excessive reduction in liver tissue in patients with notable portal hypertension, with the aim of achieving a favorable postoperative result [34,40].

Furthermore, the presence of liver stiffness measuring above 12–14 kPa can serve as an accurate marker, indicating a considerable possibility of post-hepatectomy liver failure [46]. Wu et al. determined threshold values for liver stiffness in predicting the occurrence of postoperative liver failure to be 16.2 kPa [47]. These results align with those of a prior investigation conducted by Cescon et al., which established a threshold value of 15.7 kPa and introduced the clinical utility of the preliminary assessment of liver stiffness in predicting the development of postoperative liver failure in patients with HCC. The researchers discovered that the occurrence of postoperative liver failure in individuals with HCC was 28.9%. Moreover, they observed that patients with a liver stiffness value of ≥15.7 kPa had a greater likelihood of experiencing postoperative liver failure. These findings indicate that liver stiffness, as measured by TE, may serve as a reliable approach for predicting the occurrence of postoperative liver failure in patients who undergo hepatic resection for HCC [47,48]. Additionally, Chong et al. reported a threshold value of 11.25 kPa in their study [47,49]. Another noteworthy investigation on the determination of cut-off values for the clinical benefits of liver stiffness was conducted by Rajakannu et al. The study revealed that a liver stiffness threshold of 22 kPa exhibited superior sensitivity, a negative predictive value, and a positive likelihood ratio in distinguishing patients with varying risks of hepatic decompensation when compared to an HVPG of 10 mm Hg. The aforementioned threshold assists surgeons in making clinical decisions for patients with possibly resectable HCC due to its comparable performance to the established gold standard [24,46,47]. Adding to that, it should be noted that the presence of a right lobe tumor, a high body mass index, or a limited intercostal space may compromise the accuracy of LSM measurements [26,50,51].

A significant finding is that while liver resection is a suitable treatment option for patients in stages 0 and A, according to the BCLC 2022 classification [11,12,52], our study population also included patients in stages B and C who would not typically benefit from these procedures. A considerable proportion of patients undergoing surgical intervention typically do not meet the resection criteria outlined by the BCLC due to challenges associated with the diagnosis procedure and compromised liver function [52,53,54]. However, this recurring trend is commonly observed when expanding the BCLC criteria for curative treatment. Patients in the BCLC B stage may potentially derive clinical benefits from surgical interventions, leading to a positive prognosis [52,53,54,55]. Consequently, there is a pressing need to advance research efforts aimed at developing more accurate prognostic indicators and scoring systems for survival prediction.

We settled on the above-mentioned values for HVPG and TE-LSM while taking into consideration the consensus in portal hypertension, Baveno VII [17], which states that a hepatic venous pressure gradient value equal to or higher than 10 mmHg determines the presence of CSPH in patients with viral and alcohol-related cirrhosis; in patients with non-alcoholic steatohepatitis, the threshold mentioned previously is strongly correlated with the presence of CSPH, even though signs of portal hypertension can appear at values lower than 10 mmHg. At an HVPG value above 12 mmHg, complications are likely to appear and can be associated with an even poorer outcome for patients undergoing surgical procedures [17,18,44,45]. Kim et al. concluded that the 10 and 16 mmHg cut-offs do not offer the expected information on prognosis. Hence, the implementation of a uniform HVPG cut-off across different clinical scenarios and etiologies seems overly stringent and less dependable in practical clinical settings [43]. This is exemplified by the fact that distinctions between C–P scores of 6 and 7, or 9 and 10 are not weighted enough in clinical contexts [45,56,57,58]. Multiple threshold values for LSM have been documented in the literature [12,17,45,59,60]. According to the Baveno VII consensus, hepatic transient elastography measurements below 10 kPa in the absence of any evident indications can effectively exclude the presence of compensated advanced chronic liver disease (cACLD). TE values ranging from 10 to 15 kPa are indicative of cACLD, whereas values over 15 kPa strongly support the presence of cACLD. Patients exhibiting LSM ranging from 7 to 10 kPa and experiencing persistent liver injury should be subject to individualized monitoring in order to identify any alterations that may suggest a progression toward compensated advanced chronic liver disease (cACLD). The application of the rule of five at specific thresholds (10–15-20–25 kPa) is recommended to indicate increasing relative risks of decompensation and liver-related mortality, regardless of the underlying cause of chronic liver disease [17]. In a prospective study conducted by Kim et al., patients with hepatocellular carcinoma who underwent curative LR were examined. The study indicated that LSM is an independent risk factor for post-hepatectomy liver failure. Additionally, the study established a cut-off value of 25.6 kPa for LSM [60].

Regardless of the rigorous way our study was conducted, there are several limitations. First of all, this is a retrospective study, with data being collected from medical charts. Secondly, in our institution, the HVPG assessment as a routine method for patient selection in surgical interventions has not been widely adopted in the initial stages of the retrospective research period. Along with that, there is a lack of variability among the following parameters: the C–P score and number of tumor lesions. Moreover, patients are typically diagnosed at an intermediate or advanced stage and are no longer suitable for surgical treatment, which is the main reason for the limited cohort of individuals. Another significant constraint of our research is the loss of patients, primarily due to poor follow-up adherence. Additionally, it is important to note that this study was conducted in a single medical center; therefore, it would be beneficial to replicate the study on a multi-center cohort.

## 5. Conclusions

The prognosis of hepatocellular carcinoma is heavily influenced by the underlying chronic liver disease. The predictive values in our cohort were as follows: an average MELD score of 9.16, AFP serum levels of 102 ng/mL, TBS of 4.78 cm, HVPG of 9.11 mmHg, and TE-LSM of 16.59 kPa. The majority were included in BCLC 2022 at the early stage (A-78.5%) with an average overall survival of 35.36 (±23.81) months. The MELD score showed statistically significant results in relationship with postoperative decompensation (*p* = 0.03) and 3-month survival (*p* = 0.002). While performing the Log-Rank tests to analyze the overall survival HVPG of 10 mmHg (*p* = 0.02), AFP serum levels (*p* = 0.003), and C–P classification (*p* < 0.001) proved significant in their impacts. The overall survival rates were 72.1%, 61.3%, and 21.4%, respectively, with corresponding survival durations of 69.12 (+/−6.12), 55.51 (+/−4.82), and 33.42 (+/−8.44) months, according to the low, intermediate, and high TAC scores (*p* = 0.004) influenced by AFP and C–P levels, and less by TBS.

## Figures and Tables

**Figure 1 cancers-16-01427-f001:**
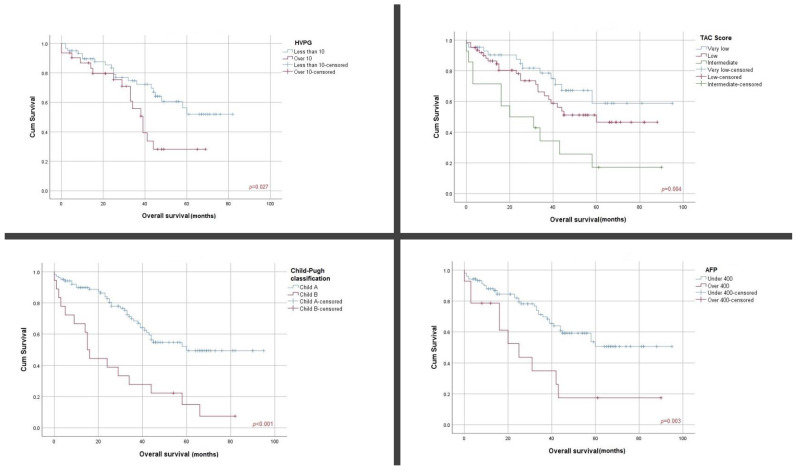
Kaplan–Meier survival curves according to HVPG, TAC score, C–P classification and AFP serum levels.

**Figure 2 cancers-16-01427-f002:**
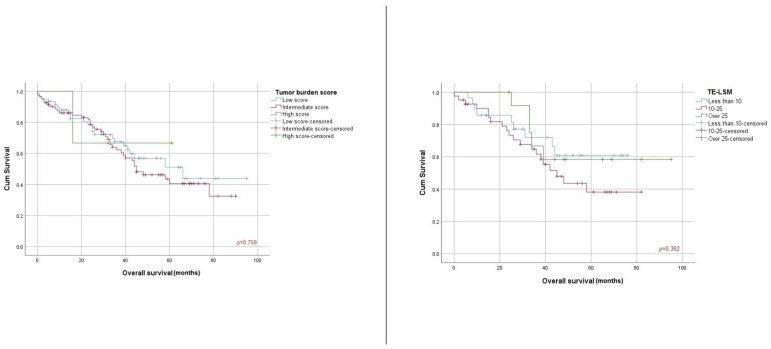
Kaplan–Meier survival curves according to TBS and TE-LSM.

**Table 1 cancers-16-01427-t001:** Clinicopathological characteristics of patients.

Parameters	Minimum	Maximum	Mean Value
Age	41	82	66.17 (±7.44)
PTD	1.3	16	4.49 (±2.58)
TBS	1	16.03	4.78 (±2.53)
HVPG	2	24	9.11 (±5.10)
TE-LSM	3.3	75	16.59 (±13.29)
DB	0.09	2.6	0.51 (±0.39)
TB	0.3	3.3	0.99 (±0.60)
ALB	2.2	5.6	4.06 (±0.67)
INR	0.92	2.5	1.20 (±0.21)
ASAT	10	425	54.21 (±57.72)
ALAT	4	588	47.72 (±59.26)
AFP	1.5	2761	102.27 (±298.81)
PLT	31	537	168.23 (±79.20)
BMI	0	3	1.85 (±0.78)
OS (months)	0	95	35.36 (±23.81)

PTD, primary tumor dimension; ALB, albumin; DB, direct bilirubin; TB, total bilirubin; ALB, albumin; INR, International Normalized Ratio; ASAT, aspartate amino transferase; ALAT, alanine amino transferase; PLT, platelet count; BMI, body mass index.

**Table 2 cancers-16-01427-t002:** Chi-squared test analyzing predictive factors for hepatocellular carcinoma.

Parameters		Postoperative Decompensation		*p* Value	3 m Survival		*p* Value	6 m Survival		*p* Value	12 m Survival		*p* Value
		No	Yes		Yes	No		Yes	No		Yes	No	
HVPG	<10	76.66%	23.34%	0.92	95%	5%	0.77	90%	10%	0.39	76.66%	23.34%	0.66
	>10	71.92%	28.08%		93.50%	6.50%		83.87%	16.13%		80.64%	19.36%	
TE-LSM	<10 kPa	82.14%	17.86%	0.68	100%	0%	0.36	96.42%	3.58%	0.07	82.14%	17.86%	0.22
	10–25 kPa	73.17%	26.83%		95.12%	4.88%		82.92%	7.08%		80.48%	19.52%	
	>25 kPa	76.92%	23.08%		100%	0%		100%	0%		100%	0%	
TBS	Low	78.33%	21.67%	0.65	95%	5%	0.76	88.33%	11.67%	0.68	78.33%	21.67%	0.65
	Intermediate	77.50%	22.50%		100%	0%		85.18%	14.82%		77.77%	22.23%	
	High	100%	0%		100%	0%		100%	0%		100%	0%	
MELD score		8.93 (±2.34)	10.06 (±3.19)	0.03	8.99 (±2.39)	11.67 (±3.96)	0.002	9.06 (±2.45)	9.79 (±3.29)	0.25	9.04 (±2.40)	9.61 (±3.14)	0.27

3 m: 3 months survival; 6 m: 6 months survival; 12 m: 12 months survival.

**Table 3 cancers-16-01427-t003:** Cox regression analysis for preoperative factors associated with overall survival.

Parameters		Univariate		Multivariate	
		OR (CI—95%)	*p* Value	OR (CI—95%)	*p* Value
TBS	Low	REF		-	
	Intermediate	1.18 (0.16–8.81)	0.86		
	High	1.42 (0.19–10.38)	0.72		
HVPG	>10 mmHg	2.08 (1.07–4.06)	**0.03**	1.92 (0.78–4.70)	0.15
TE-LSM	<10 kPa	REF		-	
	10–25 kPa	1.07 (0.35–3.20)	0.9		
	>25 kPa	1.70 (0.63–4.54)	0.28		
AFP	>400 ng/mL	2.77 (1.37–5.62)	**0.004**	12.92 (2.95–56.5)	0.001
CP	Class B	3.31 (1.84–5.95)	**<0.001**	16.17 (4.11–62.51)	<0.001
TAC score	Very low	REF			
	Low	2.98 (0.91–9.87)	**0.03**	3.81 (0.63–23.29)	0.14
	Intermediate	2.74 (0.83–9.03)	**0.04**	5.15 (1.21–21.94)	0.28

The *p* value for the univariate analysis was statistically significant for the following variables: HVPG, TE-LSM, AFP and CP. Bold = statistical significant p value.

## Data Availability

Data are fully available upon request to the corresponding authors.

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
