# Peer review of "Prognostic Indicators of Overall Survival in Hepatocellular Carcinoma Patients Undergoing Liver Resection"

_cancers, 2024, doi:10.3390/cancers16071427_

Round 1

Reviewer 1 Report

Comments and Suggestions for Authors

The present study revealed significant differences in overall survival base on  Hepatic venous 23 pressure gradient (HVPG). moreover the author identify a specific cut-off value to stratify different patients.

Overall the study can have significant impact on the hepatologist  comunity.

One suggestion to enhance the manuscript would be to expand the introduction, particularly focusing on the role of the hepatic microcirculation in chronic liver disease. This aspect holds significant importance for the conclusions drawn in this manuscript. Furthermore, the hepatic circulation plays a crucial role in regulating immune responses during fibrosis and hepatocellular carcinoma (HCC), making it a clinically relevant target. Providing a more in-depth exploration of the basic biology underlying this aspect could enrich the introduction. If the author deems it appropriate, they may consider incorporating a specific review on inflammatory cells in liver vasculature, such as the one referenced (PMID: 26188075), or similar literature, to further support and elucidate this aspect.

Author Response

Thank you for your kind review.

We improved the Introduction section - lines 78 to 88 regarding liver microcirculation alterations in relationship with HCC.

We concur that the hepatic circulation and microvascular alterations play a crucial role in modulating immune responses in fibrosis and hepatocellular carcinoma. Consequently, we went through the literature and the reference you kindly sent to supplement our work with more relevant and updated information. 

Reviewer 2 Report

Comments and Suggestions for Authors

This manuscript was aimed to evaluate prognostic indicators of overall survival in hepatocellular carcinoma patients undergoing liver resection.

I have several questions.

1.In Table 2, the percentage of hepatic decompensation with HVPG would be changed. 

2. What was the cause of mortality? Hepatic failure or HCC recurrence? They should be classified. If hepatic failure, postoperative morbidities can be described. If the HCC recurrence, recurrence free survival and Cox analysis would be added. 

3. Figure 1 requires a correction. Survival function can be deleted.

Author Response

Thank you for your kind review.

We improved the Introduction section - lines 78 to 88 regarding liver microcirculation alterations in relationship with HCC.

  1. In Table 2, there was an error, we corrected the calculated percentages of hepatic decompensation in accordance with HVPG >10 versus <10.
  2. As indicated in the manuscript, there were patients who discontinued the follow-up. Nevertheless, we possess the data pertaining to every instance of hepatic decompensation experienced by the patients throughout their hospital stay and subsequent surveillance, as well as data on recurrences. We provided in the Results section details regarding this issue - lines 228 to 235. Our research did not incorporate variables related to recurrence due to our focus on examining the overall survival of patients from several perspectives such as HVPG, TE-LSM, and TAC score. We consider that it would require another research to analyse the data on survival related to recurrences per se.
  3. We appreciate the observation. We made the correction to the figures.